

# The Cloud Feedback Model Intercomparison Project (CFMIP) Diagnostic Codes Catalogue – metrics, diagnostics and methodologies to evaluate, understand and improve the representation of clouds and cloud feedbacks in climate models

Yoko Tsushima[1], Florent Brient[2], Stephen A. Klein[3], Dimitra Konsta[4], Christine Nam[5], Xin Qu[6], Keith D. Williams[1], Steven C. Sherwood[7], Kentaroh Suzuki[8], Mark D. Zelinka[3]

[1]Met Office Hadley Centre, Exeter, United Kingdom
[2] Centre National de Recherches Météorologiques, Toulouse, France
[3] Program for Climate Model Diagnosis and Intercomparison, Lawrence Liverrmore National Laboratory, Liverrmore, USA
[4] National Observatory of Athens, Athens, Greece
[5]Universitaet Leipzig, Leipzig, Germany
[6]Department of Atmospheric and Oceanic Sciences, University of California, Los Angeles, USA
[7]Climate Change Research Centre and ARC Centre of Excellence for Climate System Science, University of New South Wales, Sydney, Australia
[8]Atmosphere and Ocean Research Institute, University of Tokyo, Kashiwa, Japan

*Correspondence to*: Yoko Tsushima (yoko.tsushima@metoffice.gov.uk)

**Abstract.** The CFMIP Diagnostic Codes Catalogue assembles cloud metrics, diagnostics and methodologies, together with programs to diagnose them from General Circulation Model (GCM) outputs written by various members of the CFMIP community. This aims to facilitate use of the diagnostics by the wider community studying climate and climate change. This paper describes the diagnostics and metrics which are currently in the catalogue, together with examples of their application to model evaluation studies and a summary of some of the insights these diagnostics have provided on the main shortcomings in current GCMs. Analysis of outputs from CFMIP and CMIP6 experiments will also be facilitated by the sharing of diagnostic codes via this catalogue.

Any code which implements diagnostics relevant to analysing clouds – including cloud-circulation interactions and the contribution of clouds to estimates of climate sensitivity in models – and which is documented in peer-reviewed studies can be included in the catalogue. We very much welcome additional contributions to further support community analysis of CMIP6 outputs.



## 1 Introduction

Cloud feedback remains the largest source of uncertainty associated with estimates of climate sensitivity using current global climate models. Evaluation of clouds is necessary not only for the assessment of model performance but also to understand how the representation of the key physical processes contributes to the errors and uncertainties. CFMIP coordinates various
experiments and the production of specific output variables to help improve our understanding of cloud-climate feedback mechanisms and processes.

To facilitate the evaluation of clouds in models using satellite observations, CFMIP has developed the CFMIP Observation Simulator Package (COSP) (Bodas-Salcedo et al., 2011). In addition, new satellite data sets have been produced which diagnose cloud properties from the observations in the same way as it is done in the simulator, e.g. by using the same criteria
for cloud detection. An example of this is the GCM-Oriented Cloud-Aerosol Lidar and Infrared Pathfinder Satellite Observation (CALIPSO) Cloud Product (GOCCP) described in Chepfer et al. (2010). This ensures that discrepancies between models and observations reveal genuine biases in the models' simulation of cloud, rather than, for example, simply highlighting differences in the definition of cloud coverage.

A range of methodologies, metrics and diagnostics have been developed, many of which utilize information on clouds
derived from the observational simulators. Use of these tools has led to considerable progress being made in understanding the uncertainties and errors associated with GCM cloud simulations over the last decade. In order for this understanding to eventually be reflected in better estimates of cloud feedbacks and climate sensitivity, it is vital to continue to develop such tools and to exploit them fully during the model development process.

To facilitate the wider use of these tools in the climate modelling community, repositories have been set up to store and
document the programs which allow their computation. The CFMIP diagnostics code catalogue lists the up-to-date repositories. Initially, a collection of repositories was set up as a part of EU Cloud Intercomparison, Process Study & Evaluation Project (EUCLIPSE) (http://www.euclipse.eu/index.html). Subsequent contributions have followed as a result of advertising the catalogue at various meetings and via the CFMIP mailing list. Other such collections of diagnostic codes have also been developed. The US CLIVAR MJO Working Group (Waliser et al., 2009) produced a collection of diagnostics
and metrics for CMIP5 which reflected shortcomings in the representation of processes that may be relevant to the simulation of the MJO (https://www.ncl.ucar.edu/Applications/mjoclivar.shtml). These scripts were applied to CMIP5 models to evaluate their simulations of various aspects of the Madden-Julian Oscillation (e.g. climate variability and predictability, see Waliser et al., 2009; Kim et al., 2009; Kim et al., 2014). The Program for Climate Model Diagnosis and Intercomparison (PCMDI) at Lawrence Livermore National Laboratory has developed common statistical error measures to
compare results from climate model simulations to observations, which have been applied to the CMIP data (e.g. Gleckler et al., 2008). This collection of well-established large- to global-scale mean climatological performance metrics provides a baseline analysis package for CMIP: The PCMDI Metrics Package (PMP) (Gleckler et al., 2016). The Earth System Model eValuation Tool (ESMValTool) (Eyring et al., 2015) has been developed for CMIP6 to enable routine comparisons of single



or multiple models, against either predecessor versions or observations. The current priority for PMP and ESMValTool is to target selected Essential Climate Variables (ECVs) (Gleckler et al., 2008; Pincus et al., 2008; Reichler and Kim, 2008), which include relative space-time root-mean square errors (RMSE) for some cloud and radiation parameters. Among the diagnostics in the current CFMIP catalogue, the Cloud Regime Error Metric for the annual mean climatology in the present

climate (Williams and Webb, 2009) has recently been included in ESMValTool.

All of this work contributes to the wider community efforts to improve the representation of models. The World Climate Research Program (WCRP) Grand Challenges represent areas of emphasis in climate science where there is believed to be likelihood of significant progress over the next 5-10 years. One requirement for these areas is to implement effective and measurable performance metrics, to build and strengthen collaborations between communities. The CFMIP diagnostics

codes catalogue will contribute to this purpose for the Clouds, Circulation and Climate Sensitivity Grand Challenge (Bony et al., 2015).

This paper is a companion paper to the CFMIP description paper for CMIP6 (Webb et al., 2017). In this paper we describe code repositories and introduce diagnostics which are currently available in GitHub repositories. A general description of the repositories is provided in Section 2. In Section 3 the individual metrics and diagnostics are described and examples of their

application to model development and understanding uncertainties are presented. Section 4 discusses some of the insights these diagnostics have provided on the main shortcomings in current GCMs, together with an outline of possible future work in this area.

## 2 General description of the repositories and the catalogue

The repositories are maintained and managed by the author of the associated diagnostic code.

In each repository, a README file and wiki page provide an introduction to the diagnostic and list the appropriate references. Different authors use different programming languages and the codes which are currently in the repository are provided in their original languages. The code repository is structured so that a version of the code in a different language can be placed in a different directory, allowing users to select the version they prefer to use. The data are assumed to be in netCDF format provided that the variables are requested by either CMIP or CFMIP. Details of each diagnostic, such as the

required input data, the outputs generated by the program and links to the source of the required observations are also provided in the repository. Basic instructions to create a repository are described in: https://github.com/tsussi/cfmip-diagnostics-code-repository.

There is no selection process for a diagnostic to be added to this catalogue. The only criterion for inclusion is that the usefulness of the diagnostic has been demonstrated in a multi-model (or multi-version) comparison in a peer-reviewed paper

and that the code repository is created in GitHub, following the instructions noted above.



## 3 Description of metrics, diagnostics and methodologies

Here we describe the diagnostics and metrics which are currently to be found in the catalogue. We start with diagnostics related to all types of clouds, followed by those focussed specifically on low-level cloud. The final group consists of diagnostics targeted at understanding cloud feedbacks. In this paper the term 'metric' refers to scalar quantities which can be easily compared to observations.

### 3.1 Simulation of ISCCP global cloud amounts (Klein et al., 2013)

https://github.com/mzelinka/klein2013-cloud-error-metrics

This is a set of four scalar measures of model fidelity in simulating clouds, where the observational "truth" is taken to be the International Satellite Cloud Climatology Project (ISCCP) cloud fields (Rossow and Schiffer 1991, 1999). These are root-mean-square differences in the space-time distributions (i.e. from 60°S to 60°N through the climatological annual cycle) of cloud variables, normalized by the space-time standard deviation of the variable from the observations. These variables are: (a) the total cloud amount for clouds with optical depth > 1.3; (b) the amount of optically intermediate and thick clouds at low, middle and high levels of the atmosphere and the impacts of optically intermediate and thick clouds on top-of-atmosphere (c) shortwave and (d) longwave radiation computed using the Cloud radiative kernels (Zelinka et al., 2012; See section 3.3 below). Figure 1 shows a summary of these measures of fidelity of CFMIP model simulations in reproducing the space-time distributions of several cloud measures, with greater fidelity indicated by smaller values, for representations of total cloud amount (a), cloud-top pressure and optical depth (b), the impacts on top-of-atmosphere shortwave (c) and longwave radiation (d). CFMIP1 models are symbols at arrow base and CFMIP2 models are symbols at arrow top. Arrows pointing to the left indicate improvement with time. The thick black arrow connects the average measure of CFMIP1 models (arrow base) to that of CFMIP2 models (arrow tip). Most individual models and the ensembles as a whole show progress over time in most measures of simulation fidelity, with small improvements for the representations of total cloud amount and large improvement for the distributions of cloud optical properties and their impact on shortwave radiation.

### 3.2 Cloud Regime Error Metric (CREM) (Williams and Webb, 2009, Tsushima et al., 2012)

https://github.com/tsussi/cloud-regime-error-metric

This is a set of scalar metrics which summarize the overall ability of a model to simulate the cloud radiative effects (CREs) of a set of different cloud regimes. The ISCCP observational cloud regimes were obtained using the KMEANS clustering algorithm (Anderberg, 1973) and an empirical method to choose the number of clusters (Rossow et al., 2005). (For details see Williams and Tselioudis, 2007.) Using daily-mean data the cloud regime assigned to a model grid box is the observational regime which has the minimum Euclidean distance in the vector space of normalized daily mean cloud top pressure, optical depth and cloud cover. Each metric is a single scalar value, so it is easy to compare different models or



different versions of the same model. These metrics can also be broken down into contributions from different cloud regimes (Eq.1).

$$M^2 = \frac{\sum_{i=1}^{n} w_i m_i^2}{n}$$

(1)

, where $n$ is the total number of the cloud regimes, $w$ is the respective area weight for the region where the regime $i$ is defined (e.g. 20°S–20°N tropics, northern hemisphere extra-tropics beyond 20°N), $m_i$ is the error in simulating the regime $i$, as defined below.

### 3.2.1 Evaluation of the annual mean climatology of cloud regimes

This is a single scalar metric which evaluates the climatological annual mean net CRE over the chosen number of cloud regimes.

The model error (RMSE) associated with each regime $i$ can be approximated with two components: the error in the relative frequency of occurrence ($f_i'$) compared to the observations ($f_i^o$) and the error in the net CRE when the regime occurs (the in-regime net CRE) ($C_i'$) compared to the observations ($C_i^o$):

$$a_i = \sqrt{(f_i' C_i^o)^2 + (f_i^o C_i')^2}$$

(2)

Figure 2 shows changes between the CFMIP-1 and CFMIP-2 models in these error components for the $a_i$ in the net CRE in the tropics (20°S–20°N). Improvements in the CFMIP2 models relative to those in CFMIP1 are seen mainly the cloud radiative properties, i.e. in the error components for the in-regime net CRE ( $f_i^o C_i'$ ), rather than those for the frequency of occurrence ( $f_i' C_i^o$ ). This is especially true for deep convective cloud, anvil cirrus, stratocumulus, transition and shallow cumulus cloud regimes.

### 3.2.2 Evaluation of the climatological seasonal cycle of cloud regimes

This scalar metric evaluates variations of climatological monthly mean net CRE over the chosen number of cloud regimes. The model error associated with each regime $i$ ($s_i$) is defined as a centred RMS error, which consists of two contributions: an error in the amplitude of the model's seasonal variation ($s_{i,amp}$) and an error in the covariance of the model's time variation for a given error in that model's amplitude ($s_{i,cov}$) (See Tsushima et al., 2013 for details). The relationship between these terms is illustrated in Fig. 3.

The seasonal variation of the shortwave CRE (SCRE) is attributable not only to the variation of clouds but also to that of the incoming solar radiation. To evaluate the variation of shortwave radiative components by clouds in models it is thus necessary to remove the latter but keep the former. This is achieved by normalizing the SCRE by the local solar insolation.

Figure 4 shows the seasonal variation of the normalized SCRE (NSCRE) of cloud regimes in the tropics (20°–20°N) (a), northern hemisphere extra-tropics beyond 20°N (b), and southern hemisphere extra-tropics beyond 20°S (c) in five CMIP5 models. The seasonal variation of the NSCRE is relatively well-simulated by the models in the tropics: the largest inter-





model spread is in the stratocumulus regime and the main differences between models relate to variations in the amplitude of the seasonal cycle.

### 3.3 Cloud radiative kernels (Zelinka et al., 2012)

https://github.com/mzelinka/cloud-radiative-kernels

The cloud radiative kernels quantify the sensitivity of the top-of-atmosphere (TOA) radiative fluxes to cloud fraction perturbations within the seven cloud top pressure categories and seven cloud optical depth categories defined by ISCCP (Fig. 5). Multiplying the cloud radiative kernels – which are a function of latitude, month, and surface albedo – by changes in cloud fraction (segregated on the same cloud top pressure-optical depth grid) between two climate states, yields a quantitative estimate of the cloud-induced TOA radiation anomalies. Normalizing these by the change in global mean

surface temperature between the two climate states then provides a measure of cloud feedback.

Because the kernels are computed using a single radiative transfer code (Fu and Liou, 1992), differences in cloud feedbacks among climate models can be unambiguously attributed to inter-model differences in the responses of clouds. Furthermore, because the cloud feedback is computed directly from changes in cloud fields rather than inferred from TOA fluxes, no adjustments are necessary to account for non-cloud-induced radiative flux anomalies. The kernels can also be applied to a

model's control simulation and observations (Klein et al., 2013, See section 3.1) to quantify radiative flux errors contributed from different cloud types in a model, or to two model versions to quantify the impact of the changes in different cloud types in a new version of the model to errors in radiative fluxes.

The panels on the right-hand side of Fig. 6 show CFMIP1 slab-ocean simulations' ensemble mean cloud radiative feedback contributions from different cloud categories in (d) longwave, (e) shortwave and (f) net, expressed per unit change in each

model's global mean surface air temperature between the two states. These estimates of cloud feedbacks are produced by multiplying the change in cloud fraction at each location and month with the collocated radiative kernels. This figure highlights the various cloud types that contribute to the cloud feedback.  High cloud changes make large but opposing contributions to the LW and SW cloud feedbacks. Low and mid-level cloud changes, which are negative in most bins, make a strong positive contribution to the SW cloud feedback, especially at optical depths greater than 3.6 where the kernel is

larger in magnitude. Because these contributions are not strongly opposed in the LW, the net cloud feedback for mid- and low-level clouds arises primarily from the SW component.

### 3.4 Zonal plots of GCM cloud and hydrometeor fraction compared with CALIPSO-GOCCP and CloudSat (Nam and Quaas, 2012)

https://github.com/chriscnam/CFMIP_LidarRadar

This code produces plots of zonally-averaged vertical profiles of cloud fraction and hydrometeor fraction from the active-sensor satellite observations and models. Complementing the ISCCP simulator referred to above, the active lidar and radar satellite simulators emulate the radiances which would be retrieved by the CALIPSO and CloudSat instruments within



climate models. The active lidar and radar simulators allow a more accurate comparison of the vertical distribution of clouds and hydrometeors in climate models with the CALIPSO-GOCCP and CloudSat 2B-GeoProf datasets (Marchand et al., 2009). Figure 7 shows the zonally-averaged cloud fraction (top row; Fig. 7 a-c) and hydrometeor fraction (bottom row; Fig. 7 d-e) for June-July-August 2007 from: (a) CALIPSO-GOCCP data; (b) IPSL5B GCM with COSP Lidar Simulator; (c) IPSL5B GCM; (d) CloudSat data; and (e) IPSL5B GCM with the COSP Radar Simulator (Nam and Quaas, 2012; Chepfer et al.,2010, Marchand et al., 2009). From these plots, one can identify model biases such as the overestimate of optically thin high-level clouds and the significant underestimate of mid-level and (sub) tropical low-level clouds. In addition, it can be seen that IPSL5B overestimates the frequency of precipitation. These findings imply that compensating mechanisms in IPSL5B balance out the radiative imbalance caused by incorrect optical properties of clouds and consistently large hydrometeors in the atmosphere, which was also found for ECHAM5 in Nam and Quaas (2012).

### 3.5 A-train satellite instantaneous cloud property observations for process-oriented evaluation (CALIPSO-PARASOL) (Konsta et al., 2015)

https://github.com/dimitrakonsta/process-oriented-cloud-evaluation

These 2D histograms correlate different cloud variables from the multi-sensor A-train observations at the instantaneous time scale, and at high spatial resolution. This allows us to see how different key cloud properties vary as a function of one another (Konsta et al., 2012) and to build pictures of cloud processes which are well suited for the evaluation of clouds in climate models.

Specifically, the histogram shows the relationship between cloud fraction and the cloud vertical distribution from CALIPSO (Winker et al., 2007) and cloud reflectance measured by PARASOL (Parol et al., 2004), which is a good surrogate of the cloud optical depth. The same relationship is reproduced for the model using the COSP simulator.

Figure 8 provides the instantaneous (*upper panels*) and the monthly mean (*lower panels*) relationship between the cloud cover and the cloud reflectance over the tropical oceans for the observations (a, d) and as simulated by two versions of the LMDZ5 model using COSP (b, c, e, f). In the observations, the cloud reflectance tends to increase with increasing cloud cover, but the models do not reproduce this relationship in either the instantaneous data or the monthly mean. However, the two panels show that the relationship between cloud fraction and cloud optical thickness is very different when using monthly mean and instantaneous values, both in observations and in the models. This joint data analysis of different instantaneous observations allows for a more precise evaluation of clouds properties and model parameterizations.

### 3.6 Low-level cloud distribution and optical properties: CALIPSO, Parasol, CERES (Nam et al., 2012)

### 3.6.1 Vertical distribution of low-level clouds

https://github.com/chriscnam/CFMIP_LowCloudDistribution

These histograms show the frequency of occurrence of clouds below 4 km over the tropical oceans (30°N–30°S) in the observations and models (Fig.9). This diagnostic, from Nam et al. (2012), identifies non-overlapped (i.e. by mid- and high-





level cloud) low-level clouds within subsiding regimes. This is done by first identifying where large-scale vertical velocities at 500 hPa and 700 hPa are greater than 10 hPa day$^{-1}$; then distinguishing between shallow cumulus and stratocumulus regimes using the lower tropospheric stability threshold of 18.55 K, as defined in Medeiros and Stevens (2011); and finally testing whether lidar-defined high- and mid-level cloud cover are both less than 5 %. The histogram demonstrates that CMIP5 models tend to concentrate their low-clouds in the lowest 1 km of the troposphere, regardless of the large-scale environment, instead of distributing them through out the boundary layer.

### 3.6.2 Shortwave cloud radiative effect (SW CRE) and Parasol reflectance

https://github.com/chriscnam/CFMIP_SWCRE_Parasol

The poor vertical distributions of low-level clouds shown above lead to biases in their optical properties, which can be quantified using this diagnostic: the mean SW CRE and Parasol reflectance broken down into different cloud fraction bins (Fig. 10). The comparison of short-wave cloud radiative effect from various CMIP5 models with CERES TOA fluxes as well as Parasol reflectance above the non-overlapped low-clouds  (Fig. 10) shows that models overestimate the cloud radiative effects compared to observations, even for comparable cloud fractions and large-scale environmental conditions (Nam et al., 2012).

### 3.7 Warm rain microphysical process diagrams (Suzuki et al. 2015)

https://github.com/kntrszk/cfodd

This diagnostic plots vertical profiles of radar reflectivity in the form of a contoured frequency diagram as a function of in-cloud optical depth (ICOD). The radar reflectivity is obtained from the CloudSat 2B-GEOPROF product (e.g. Marchand et al. 2008) and the cloud optical depth is obtained from the MODIS cloud product (e.g. Platnick et al. 2003; Nakajima et al. 2010a). The diagram is constructed from the probability density function (PDF) of radar reflectivity at each ICOD bin, and shows the PDFs as contoured frequency of radar reflectivity as a function of ICOD, which is referred to as the Contoured Frequency by Optical Depth Diagram (CFODD). The ICOD is determined by a vertical slicing of the total cloud optical thickness from MODIS into each radar bin according to the adiabatic growth assumption that provides a scaling function of the optical depth with respect to geometric height. When the statistics thus constructed are further classified according to the cloud-top particle size also obtained from MODIS (Platnick et al. 2003; Nakajima et al. 2010a), the vertical microphysical structure and its microphysical transition from non-precipitating through drizzling to raining are clearly depicted (Suzuki et al. 2010; Nakajima et al. 2010b), as shown in Fig. 11 (top panel). Corresponding statistics are also constructed from model output of satellite cloud observables obtained with the aid of appropriate satellite signal simulators such as CFMIP Observation Simulation Package (COSP; Bodas-Salcedo et al. 2011) and Satellite Data Simulation Unit (SDSU; Masunaga et al. 2010). The model-derived statistics thus synthesized (Fig. 11 lower panels) are then compared with the satellite-based statistics to identify key model biases in representing warm rain formation process characteristics. Examples for such an analysis with two CMIP5 models, shown in Fig. 11, demonstrate how some models share a common bias of "too early rain



formation" that happens even when the cloud-top particle sizes are small, in stark contrast to the satellite statistics (Suzuki et al. 2015).

**3.8 Sensitivity of Tropical Low-Cloud Reflection to surface temperature change at various time scales (Brient and Schneider, 2016)**

https://github.com/florentbrient/Cloud-variability-time-frequency

https://github.com/florentbrient/ECS-Constraint

This diagnostic calculates covariances of time series of any cloud-related variable (e.g. low-cloud albedo $\alpha_c$, low-cloud fraction) and that of sea surface temperature $T$ (robust regression slope, correlation coefficients).

Brient and Schneider (2016) estimated the sensitivity of the reflectance of tropical low clouds (TLC) to the underlying surface temperature change at intra-annual, seasonal and interannual timescales in the observations and CMIP5 models. As shown in Fig. 12, they found that in the observations on all timescales shortwave reflection by TLC decreases robustly when the underlying surface warms. They also showed that in simulations of the warmer climate reached after quadrupling carbon dioxide concentrations higher sensitivity (HS) models project a reduction of TLC reflection, whereas lower sensitivity (LS) models project less change or even an increase. The models' equilibrium climate sensitivity (ECS) correlates significantly with the sensitivity of cloud reflection to the underlying surface temperature warming (53% of the variance). Additionally, the spread in temporal covariance of low-cloud reflection with surface temperature in current climate simulations explains about half of the ECS variance across models. Therefore, recent space-based measurements of short-wave radiation permit estimation of a more likely range of ECS values, highlighting that ECS values below 2.3 K appear very unlikely. Code is provided to calculate a set of diagnostics based upon the covariance of two time series (robust regression slope, correlation coefficients), separation of time scales is achieved by applying low-pass and high-pass filters. The calculation by the code also provides the uncertainty range underlying the covariance through a stationary bootstrap procedure. An additional program is used to constrain the more likely range of ECS values by weighting models' ability to reproduce the observed covariance between TLC reflection and sea surface temperature.

**3.9 Sensitivities of low cloud cover to estimated inversion strength and sea surface temperature (Qu et al., 2014)**

https://github.com/xinqu2016/SST-and-EIS-slopes

This metric calculates the sensitivity of tropical marine low cloud cover (LCC) to two key cloud-controlling factors, the strength of the inversion capping the atmospheric boundary layer (measured by the estimated inversion strength, EIS) and sea surface temperature (SST). These parameters were developed as part of a heuristic model used to interpret change in LCC simulated in GCMs. The heuristic model's premise is that simulated LCC changes can primarily be interpreted as a linear combination of contributions from EIS and SST. For a given GCM, the respective contributions of EIS and SST are computed by multiplying (1) the sensitivity of LCC to EIS and SST variations, by (2) the climate-change signal in EIS or



SST. The heuristic model is remarkably skillful, capturing a large portion of the variance of LCC changes across different GCMs. In particular, its SST term dominates, accounting for much of the spread in simulated LCC changes.

The sensitivities of LCC to SST and EIS (referred to as the SST and EIS slopes, respectively) were computed based on interannual variability in the 20th century via multiple regression analysis and for each of five low cloud-dominated oceanic

regions. Figure 13 shows the EIS slope ($\partial LCC / \partial EIS$) and the SST slope ($\partial LCC / \partial SST$) in 36 CMIP3 and CMIP5 models and the observations. According to the observations, slopes are negative for SST and positive for EIS. This suggests that LCC decreases with increasing SST, but increases with increasing EIS. Model-simulated slopes generally have signs consistent with the observations but underestimate the magnitudes of both SST and EIS slopes. The observational slopes were computed based on ISCCP cloud data (Rossow and Schiffer 1991), ERA-Interim reanalysis (Dee et al. 2011) and

NOAA optimum interpolation monthly SST version 2 (Reynolds et al. 2002) during the period 1984-2009.

## 3.10 Lower Tropospheric Mixing Indices (Sherwood et al., 2014)

https://github.com/scs46/LTMI-mixing

The Lower Tropospheric Mixing Index (LTMI) proposed by Sherwood et al. (2014) was found to be empirically related to climate sensitivity in both CMIP3 and CMIP5 models. The mixing diagnosed via this index is intended to capture vertical

mixing not directly associated with precipitation production, such that the LTMI can also be interpreted as a measure of bulk precipitation inefficiency. Sherwood et al. (2014) argued that the relationship seen between LTMI and climate sensitivity arises because a high LTMI implies that upward moisture fluxes within the troposphere will increase relatively strongly with temperature, producing more positive global net cloud feedback by inhibiting the conditions necessary for low cloud formation. The LTMI consists of two components, which represent two scales of vertical mixing: small-scale vertical

mixing (S) within a single grid-column of the model, involving model parameterizations directly, and large-scale mixing (D) via explicitly resolved, shallow overturning circulations. S is diagnosed from vertical gradients of humidity and temperature over warm tropical oceans at altitudes around the typical marine boundary-layer top, while D is calculated explicitly from model pressure velocity fields in the lower and middle troposphere. Both quantities were obtained from annual-mean data over tropical oceans, with only two years giving reasonably stable results compared to the large differences in mixing rates

between models, although we recommend longer time periods. Figure 14 shows the relationship of ECS to S, D and the LTMI (the sum of S and D). Observations of S and D were obtained in that study from radiosonde and reanalysis data. The ranges of D and S are similar (Fig. 14a, b), and the LTMI explains about 50% of the variance in total system feedback (r=0.70) and ECS (r=0.68) (Fig. 14c), thus LTMI explains a significant portion of the model spread. In the observations, S shows near the middle of the GCM range, but D close to the top end, which suggests the existence of mid-level outflows

stronger than models.



### 3.11 Application to understanding and model development

Here two examples are presented of how the metrics and diagnostics described in the previous section are applied to models during their development.

Bodas-Salcedo et al. (2012) applied cloud regime analysis in this catalogue (Section 3.2) diagnosed cloud regimes around
cyclone centres over the Southern Ocean in observations and in an atmospheric-only configuration (GA2.0) of the Met Office Unified Model. The motivation for this study was to investigate the role of clouds in the long-standing bias of surface downwelling shortwave radiation over the region. They found that low- and mid-level clouds in the cold-air sector of the cyclones are responsible for most of the bias (Fig.15). Based on this analysis, a new diagnosis of shear-dominated boundary layers was developed and was included in a newer configuration of the model.

Kamae et al. (2016) applied two diagnostics in this catalogue to a Multi-Parameter Multi-Physics Ensemble (MPMPE), which consists of both parametric and structural uncertainties in parameterizations of cloud, cumulus convection and turbulence to investigate relationships between the LTMI and equilibrium climate sensitivity (Section 3.11). Significant correlations were found in all of the perturbed parameter ensembles (PPEs) with different physics schemes but using an old
convective scheme, but not in PPEs which used a new convective scheme. To understand the difference they used the Cloud radiative kernels (Section 3.3) and broke down the differences into contributions from different cloud types. In both subset of ensembles with different convective schemes they found significant positive correlations between small-scale mixing ($M_{small}$) and low-level cloud shortwave feedback ($\lambda_{SWcld}$), i.e. the larger the mixing the more positive the cloud feedback. Although middle-level cloud shortwave feedbacks also have significant correlations with $M_{small}$ in both ensembles, their signs of the
correlation are the opposite, negative in the PPEs with the new convective scheme and positive in the PPEs with old convective scheme (Fig. 16). In the PPEs with the new convective scheme, the relationships of low-level cloud feedback and middle-level cloud feedback to small-scale mixing are opposite, hence cancel each other. As a result, the climate sensitivity has no significant correlation to the LTMI. They suggest that a different mechanism other than lower tropospheric mixing could control middle-level cloud feedback and there is therefore a need to develop an alternative emergent constraint.

### 4 Discussion

We have described the metrics and diagnostics that are currently available in the CFMIP diagnostics code catalogue and have provided examples of their application to model evaluation. These examples demonstrate value of these diagnostics in understanding and reducing errors in the representing clouds in climate models.

We envisage the metrics and diagnostics in this catalogue being used extensively for model evaluation studies in CMIP6,
particularly as part of CFMIP. The ISCCP cloud histograms defined in terms of cloud top pressure and cloud optical thickness have been used to understand both model errors and feedbacks in different cloud types and regimes (Klein et al., 2013, Williams and Webb, 2009, Tsushima et al., 2013, Tsushima et al., 2015, Zelinka et al., 2012). Some of these studies



use instantaneous, i.e. time-step, data (e.g. Konsta et al., 2015, Suzuki et al., 2015), the motivation being to understand physical processes, as this is known to be important for understanding cloud feedbacks (e.g. Gettleman and Sherwood., 2016).

As the spread in low cloud feedbacks in the tropics was a large contribution to the spread in climate sensitivity in both IPCC
AR4 (Randall et al., 2007) and IPCC AR5 (Boucher et al., 2013), many studies have focussed on the representation of low clouds and their associated feedbacks in climate models. Indeed, about half of the diagnostics in the current catalogue are targeted at low-level clouds (Nam et al., 2012, Qu et al., 2013, Sherwood et al., 2014, Brient and Schneider 2016, Suzuki et al., 2015). These studies have provided insights into three long-standing problems in GCMs.

a) The too few, too bright problem. Low clouds in GCMs have larger reflectance but smaller cloud amount than the
observations. Cloud reflectance sorted by cloud amount showed that models overestimate the cloud radiative effects compared to observations, even for comparable cloud amounts (Nam et al., 2012). Deficiencies which were highlighted in Nam et al. (2012) to explain the overestimate of low cloud radiative effect are: misrepresentation of horizontal inhomogeneity of cloud optical properties and the vertical overlap of cloud layers. In addition, 3D effects have a significant impact on the solar reflection and a fast algorithm to account for this in global atmospheric models is being developed
(Hogan et al., 2016). Tsushima et al. (2015) confirmed the overestimate of in-cloud albedo by comparing daily ISCCP cloud regime data with the regimes simulated in five current models. In the stratocumulus regime, models simulate smaller cloud amounts than those observed because broken cloud situations tend to occur more frequently than overcast situations, in contrast to the observations. The too frequent occurrence of broken clouds contributes more to the positive bias in reflectance for the stratocumulus regime than the overestimate in reflectivity for a given cloud cover. Further investigation of
the reasons for the underestimate of overcast cases in models is necessary.

b) Vertical profile of low-level clouds. Nam et al. (2012) showed that GCMs poorly represent the vertical structures of low-level clouds. Sherwood et al., (2014) showed lower tropospheric mixing in GCMs is smaller than in observations and suggest that most models underestimate climate sensitivity. The mixing consists of small-scale mixing by convective and other parameterizations (e.g. Brient et al., 2016) and large-scale mixing by large-scale circulations. Whether this mixing is
achieved in practice in a model will depend upon the precise details of the convection scheme, in particular the model's ability to represent shallow convection.

CFODDs (Suzuki et al., 2015) diagnose the cloud-rain conversion process which is related to the vertical distribution of lower tropospheric humidity. These diagrams show that the formation of rain from cloud droplets in GCMs (often referred to as auto-conversion) happens much faster than it does in the observations. In addition, convection schemes in GCMs tend to
convert extra moisture into precipitation immediately. Zhao et al. (2016) used a developmental version of the next-generation Geophysical Fluid Dynamics Laboratory GCM and constructed a tightly controlled set of GCMs where only the formulation of convective precipitation is changed. They demonstrated that model estimates of climate sensitivity can be strongly affected by the rain formation process in a model's convection parameterization. The model differences are dominated by shortwave feedbacks and come from broad regimes ranging from large-scale ascent to subsidence regions.



Better representation of these processes could therefore help to improve simulations of the vertical profiles of lower tropospheric humidity and clouds.

c) Low cloud amount change with SST increase

Both GCMs and process models tend to produce positive low-cloud feedbacks through a reduction of low cloud amount. However, deficiencies in the representation of low clouds in GCMs, as well as a lack of observational constraints, means that the sign of the low-cloud feedback is still very uncertain (Boucher et al., 2013). Positive low cloud feedback in the observations in all timescales was shown by Brient and Schneider (2016). Qu and Hall (2014) showed that interannual variations of low cloud cover decrease with increasing SST in the observations, and confirmed that models tend to reproduce this decrease in both historical and climate change simulations. They also found that inter-model variance of low cloud changes in climate change simulations is dominated by the inter-model differences in the SST increase and the sensitivity to SST. Why then does low cloud amount decreases with increasing SST in GCMs? The mechanism proposed by Sherwood et al. (2014) is that intensification of lower tropospheric mixing could dry the boundary layer and reduce cloud amount. These observational constraints of low cloud amount feedback suggest larger positive cloud feedback and hence higher climate sensitivity. These diagnostics, which identify a source of error in GCMs that relates to climate predictions, merit attention from those developing climate models and climate observations (Klein and Hall, 2015). The possible contributions of other factors to the low cloud cover change should also be examined (e.g. Webb and Lock, 2013). In Large-eddy Simulations (LES) at stratocumulus locations, the cloud remains overcast but thins in the warmer, moister, $CO_2$-enhanced climate, due to the combined effects of an increased lower-tropospheric vertical humidity gradient and an enhanced free-tropospheric greenhouse effect that reduces the radiative driving of turbulence (Bretherton et al. 2013). Mechanisms of low-level cloud amount change in warming climate are still not well understood and further investigations combining observations, GCMs and process models are necessary.

Although there is a significant correlation between LTMI and ECS in both CMIP3 and CMIP5 models, its correlation to cloud radiative feedback is weaker. Kamae et al. (2016)'s investigation of the lower tropospheric mixing using MPMPEs found that small-scale mixing has a significant correlation with low-level cloud shortwave feedbacks and also that the sign of the correlation is robust across the ensembles. Although correlations were also found with middle-level cloud shortwave feedback, the signs are not robust among the different physics ensembles. Zelinka et al. (2012) showed that high cloud changes induce wider ranges of LW and SW cloud feedbacks across models than do low clouds. Zhao et al. (2016)'s study suggests that changes in convective clouds may be as important as those in low clouds in determining climate sensitivity. Hence development of diagnostics and emergent constraints associated with different cloud types and processes would be helpful.

Underestimation of middle-top clouds has been a common bias in climate models (Zhang et al., 2005, Tsushima et al., 2014), but its implication for cloud feedbacks is currently not well understood. Watanabe et al. (2012) found that MIROC5 underestimates middle-top clouds much less than MIROC3 and that the cloud feedback in MIROC5 is much less positive



than in MIROC3. One of the main reasons for this is an increase in middle-top clouds in response to global warming in MIROC5. Greater understanding of middle-level clouds and their associated feedbacks will be useful.

For high clouds, the Fixed Anvil Temperature (FAT) mechanism (Hartmann and Larson, 2002) suggests that the temperature of the detrained tropical anvils associated with deep convection remains unchanged in a warming climate, implying that the cloud altitude feedback is positive. This mechanism, however, does not explain whether the cloud amount will increase or decrease. In addition, high thin cirrus which spreads into the Tropical Transition Layer may be associated with different feedback mechanisms. High cloud amount tends to decrease in current conventional GCMs (Zelinka et al., 2012) but a global cloud resolving model shows an increase (Tsushima et al., 2014), suggesting that the response could be dependent on certain parameterization schemes, in particular convection and microphysics. Further evaluation of high clouds and examination of possible high cloud amount feedback mechanisms will clearly be necessary. Radar and Lidar reflectivity-height histogram from CloudSat and CALIPSO were used to evaluate cloud amount and vertical profiles of high clouds (Kodama et al., 2012, Williams et al., 2015). Histograms such as these and other diagnostics using this data should be useful for this work.

With regard to physical processes, changes in thermodynamic phase in some clouds are expected in warming climate (Senior and Mitchell, 1993). Tsushima et al. (2006) highlighted the importance of evaluating the ratio of ice and liquid in mixed-phase clouds in simulations of the current climate because it determines how large the phase change might be under climate change. An underestimate of the relative amount of supercooled liquid water has been found in GCMs (Cesena et al., 2014, Tan et al., 2015). Tan et al. (2015) demonstrated that, as a consequence of the larger increase in liquid water from excessive ice water in the control climate, models could underestimate climate sensitivity. Bodas-Salcedo et al. (2016) used a cyclone-composite technique, and quantified the contribution of different regions around cyclone centres to the solar radiation budget and the feedback over the Southern Ocean. These methodologies and diagnostics could be useful for evaluating mixed-phase clouds in models.

Understanding clouds and cloud feedbacks as a part of dynamical systems and their response to climate change will also be important. Some dynamical responses are known to be robust among GCMs, such as the expansion of the Hadley Cell (e.g., Seidel et al., 2008; Johanson and Fu, 2009) and the poleward shift of the mid-latitude jets (e.g., Yin, 2005; Yu et al., 2010; Barnes and Polvani, 2013). Grise et al. (2016) investigated the impact of these dynamical responses to climate sensitivity using CMIP5 models and found that in the Southern Hemisphere inter-model differences in the value of ECS explain ~60% of the inter-model variance in the annual-mean Hadley cell expansion but just ~20% of the variance in the annual-mean midlatitude jet response. Tselioudis et al. (2016) investigated the relationship between interannual variations of the latitudinal position of clouds and their radiative effects to those in the Hadley cell and the mid-latitude jets. They found that the interannual variations of the locations of high clouds and the Hadley cell are correlated significantly but did not find a robust correlation between clouds and the mid-latitude jets. Development of diagnostics which evaluate the representation of clouds within the major large-scale dynamical systems and their variations will therefore be useful. Metrics that explicitly



include measures of circulation or water vapour and their relationships to clouds (e.g. the vapour-cloud relationships described by Bennhold and Sherwood, 2008) are likely to aid the understanding of cloud errors in models.

This paper describes only those emergent constraints which are currently included in the catalogue. Various emergent constraints for ECS have been proposed using both CMIP3 and CMIP5 models (Klein and Hall, 2015) and more will

undoubtedly be developed in the future. Development of emergent constraints for climate sensitivity or particular climate feedbacks which are underpinned by clear hypotheses and related to physical processes will be required.

We anticipate that the CFMIP diagnostic codes catalogue will continue to expand and invite additional contributions to further support the community to analyze CMIP6 outputs and to help develop and improve our understanding of cloud

processes and cloud feedbacks.

**Code and Data Availability**

Each repository is linked to the CFMIP webpage and can be found in the diagnostics code page there: https://www.earthsystemcog.org/projects/cfmip/. The following page also has links to metrics that are included in the

catalogue: https://github.com/tsussi/cfmip-diagnostics-code-repository.

CMIP data are available through the PCMDI CMIP page: http://www-pcmdi.llnl.gov/projects/cmip/

CFMIP1 and CFMIP2 data can be found under CMIP3 CMIP5, respectively.

**Acknowledgements**

We are grateful to Drs. Mark Ringer and Gill Martin for helpful comments on the manuscript. This work was supported by the Joint UK BEIS/Defra Met Office Hadley Centre Climate Programme (GA01101). This work was originally funded by the European Union, Seventh Framework Programme (FP7/2007-2013) under grant agreement no 244067 via the EU Cloud Intercomparison and Process Study Evaluation Project (EUCLIPSE). X. Qu, M.D. Zelinka and S.A. Klein are supported by

the United States Department of Energy's Regional and Global Climate Modelling Program under the project "Identifying Robust Cloud Feedbacks in Observations and Models". The work of M.D. Zelinka and S.A. Klein was performed under the auspices of the United States Department of Energy by Lawrence Livermore National Laboratory under contract DE-AC52-07NA27344.



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



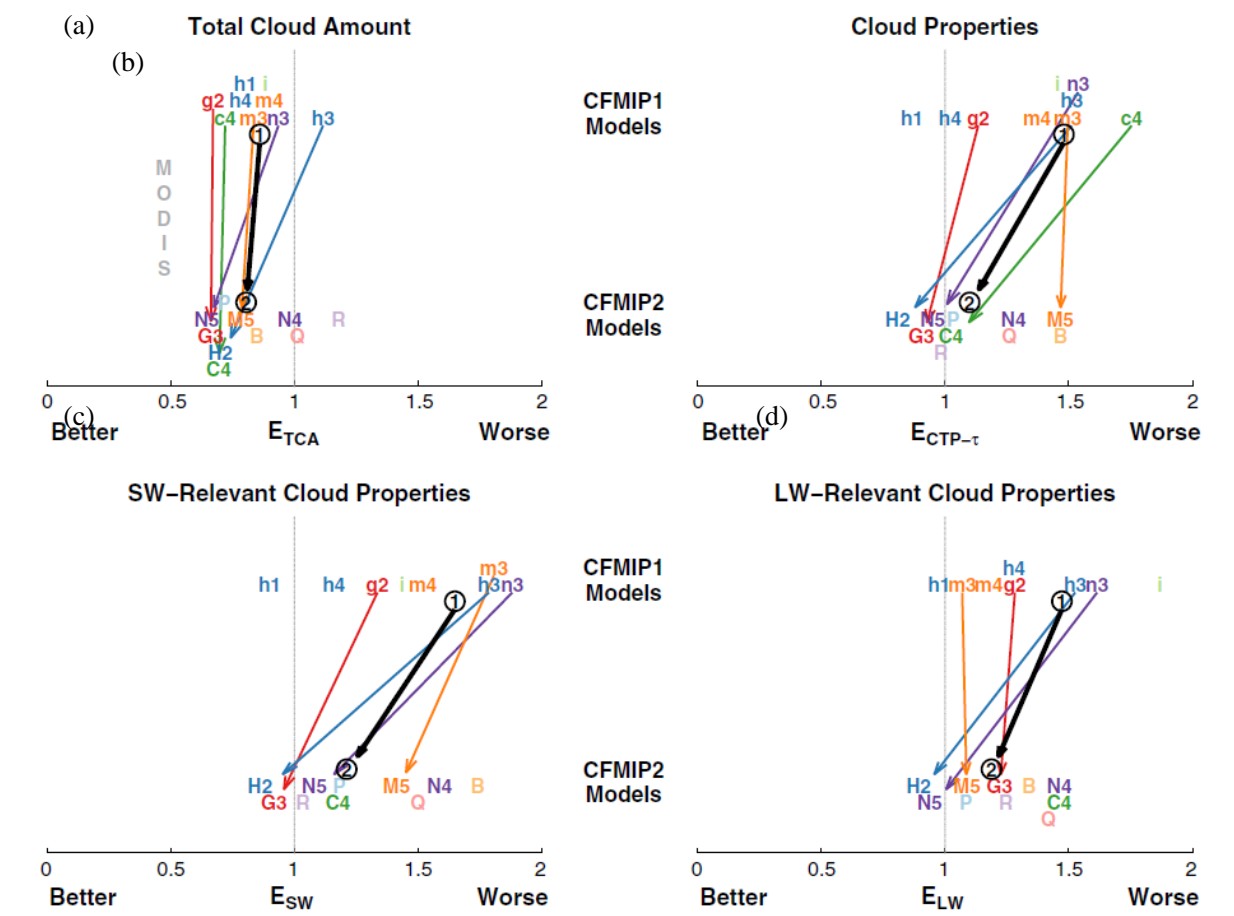

**Figure 1.** Scalar measures of fidelity of CFMIP model simulations in reproducing the space-time distribution of several cloud measures, with greater fidelity indicated by lower $E$ values. $E_{TCA}$ (a) measures fidelity in simulating total cloud amount, whereas $Ectp$-$\tau$ (b) measures fidelity in simulating cloud-top pressure and optical depth in different categories of optically intermediate and thick clouds at high, middle, and low-levels of the atmosphere. The impacts on top-of-atmosphere shortwave and longwave radiation in the same categories used for $Ectp$-$\tau$ are measured by $E_{SW}$ (c) and $E_{LW}$ (d), respectively. Models are stratified vertically into the two ensembles and are plotted in different symbol keys. (To identify a model in a symbol key, see Klein et al., 2013.) For the modeling centers in which we can track progress, the arrow connects the oldest model in the family (arrow base) to the most recent model (arrow tip). The thick black arrow connects the average measure of CFMIP1 models (arrow base) to that of CFMIP2 models (arrow tip). Arrows pointing to the left indicate improvements with time. Reprinted from Klein et al. (2013).



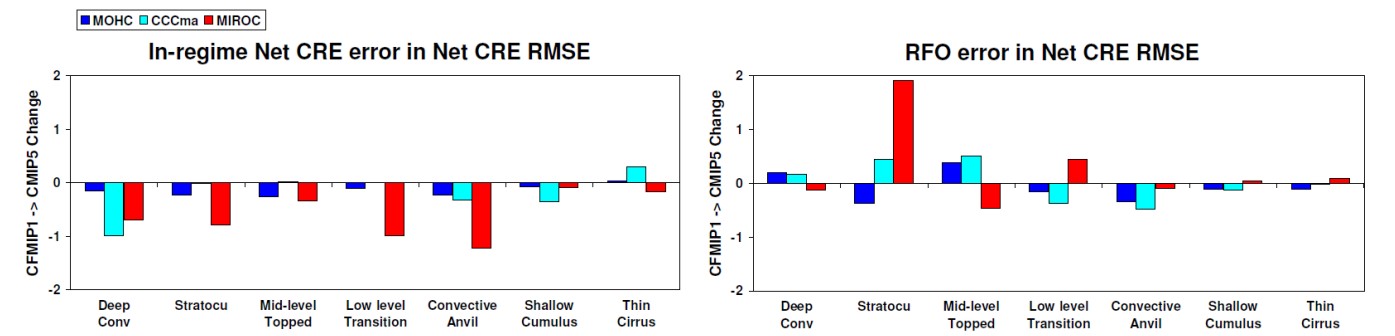

**Figure 2.** Changes between CFMIP-1 and CFMIP2 of RMSE components of $a_i$ for net CRE within daily ISCCP simulator cloud regimes in the tropics. (a) In-regime net CRE components ($f_i^o C_i^{'}$), (b) frequency of occurrence components ($f_i^{'} C_i^0$). Cloud regimes are in the order of larger albedo. A graph drawn using the values in Table 2 in Tsushima et al. (2013).





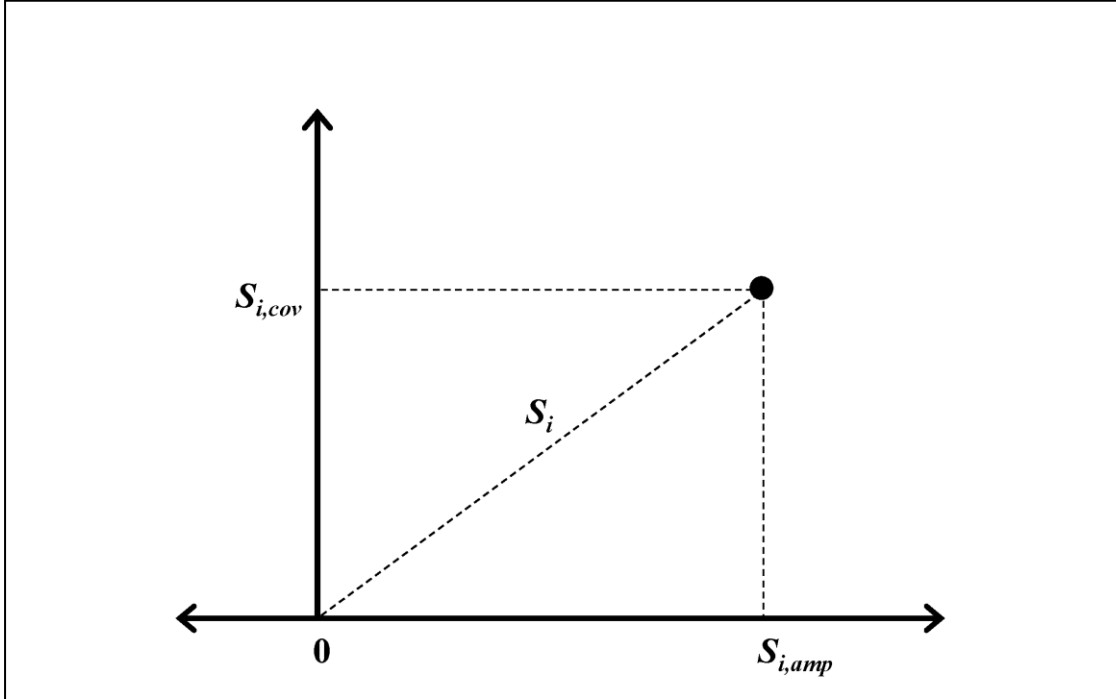

**Figure 3.** Geometric explanation of the RMS error diagram. The x-axis corresponds to the contribution of amplitude error to the RMS error; $s_{i,amp}$ while the y-axis corresponds to the contribution from pattern error in the time variation to the RMS error; $s_{i,cov}$. Redrawn from Tsushima et al. (2013).





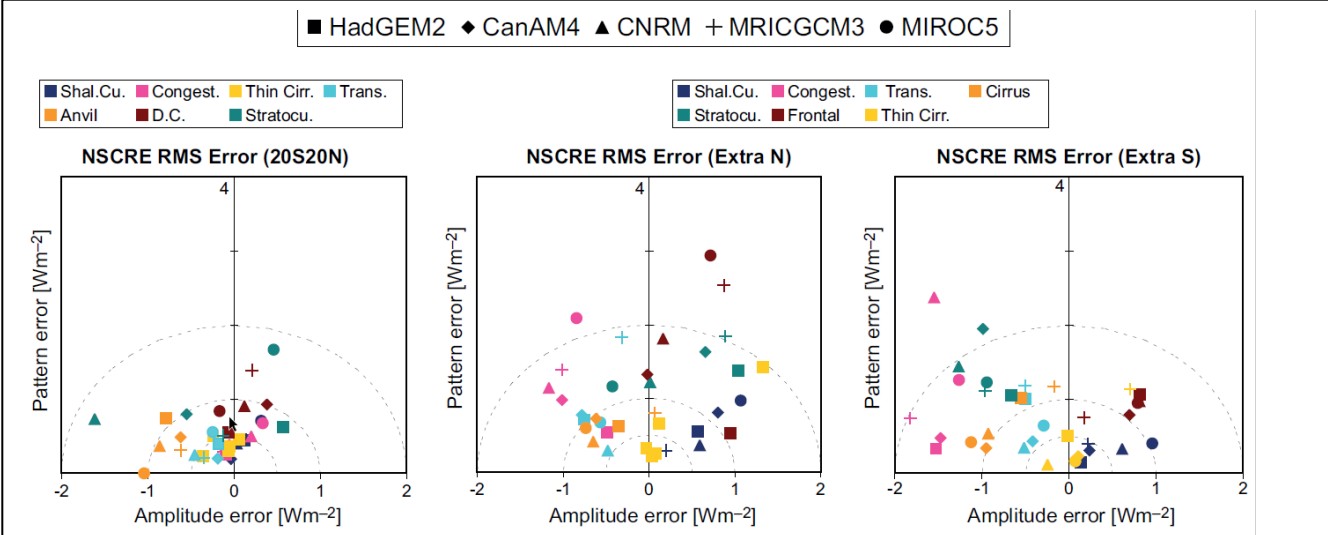

**Figure 4.** Centred RMS error diagrams of the seasonal variation of NSCRE of cloud regimes in (a) the 20°S - 20°N, (b) northern extra-tropics beyond 20°N, and (c) southern extra-tropics beyond 20°S. Colours distinguish cloud regimes. Marks distinguish models. The dotted lines are contours of the magnitude of $s_i(NSCRE)$. The x-axis shows the contribution of amplitude error while the y-axis shows the contribution from pattern error in the time variation. Reprinted from Tsushima et al. (2013).





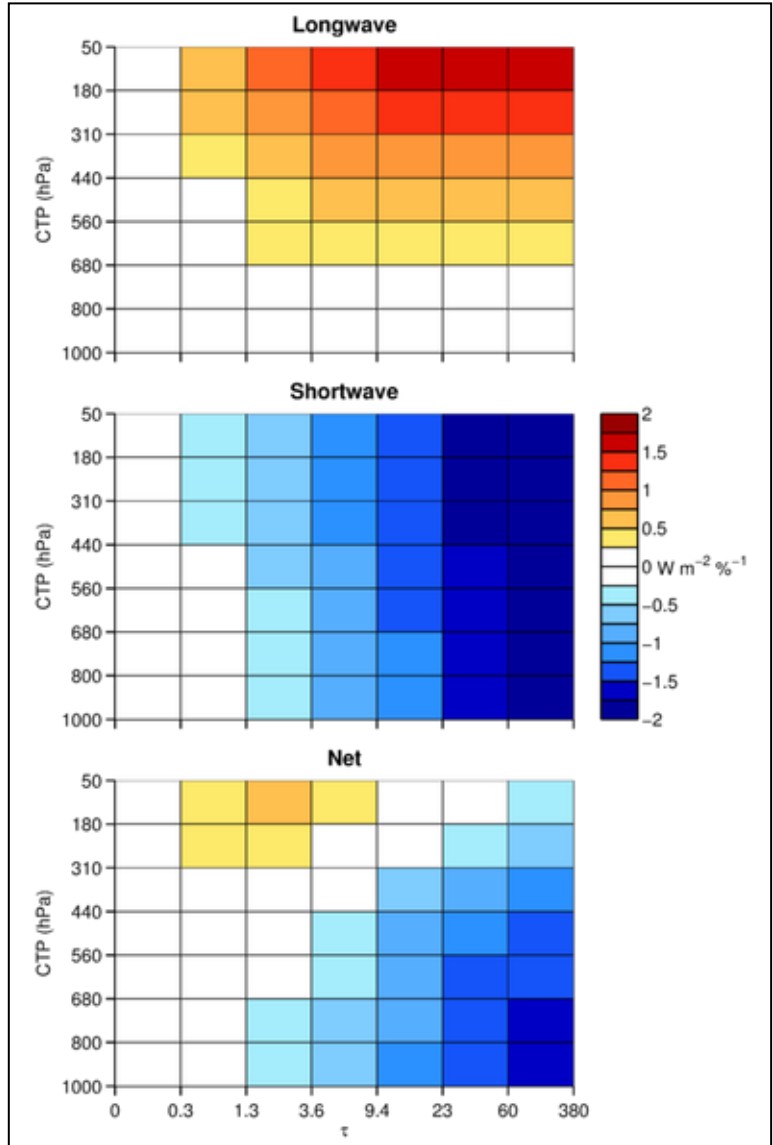

**Figure 5.** Global, annual, and ensemble mean (a) LW, (b) SW, and (c) net cloud radiative kernels. In each model, the kernels have been mapped to the control climate's clear-sky surface albedo distribution before averaging in space; thus, the average kernels are weighted by the actual global distribution of clear-sky surface albedo in each model. Redrawn with modification from Zelinka et al. (2012).





**Figure 6.** Global, annual, and ensemble mean cloud fraction for the (a) $1 \times CO2$ and (b) $2 \times CO2$ runs, along with (c) the average difference expressed per unit change in each model's global mean surface air temperature between the two states. Matrix resulting from multiplying the change in cloud fraction at each location and month with the collocated (d) LW, (e) SW, and (f) net cloud radiative kernels, then taking the global, annual, and ensemble mean. The sum of each matrix is shown in each title. Bins containing an ''×'' indicate those in which ≥75% of the models agree on the sign of the field plotted. Reprinted from Zelinka et al. (2012) ©American Meteorological Society. Used with permission.





**Figure 7.** Zonal cloud and hydrometeor fraction for JJA 2007. Cloud Fraction (Top Row): (a) CALIPSO-GOCCP data, (b) IPSL5B with CALIPSO simulator, (c) IPSL5B cloud fraction. Hydrometeor Fraction (Bottom Row): (d) CloudSat data, (e) IPSL5B with CloudSat simulator. Produced for IPSL5B with the observational data used in Nam and Quaas (2012) ©American Meteorological Society. Used with permission.



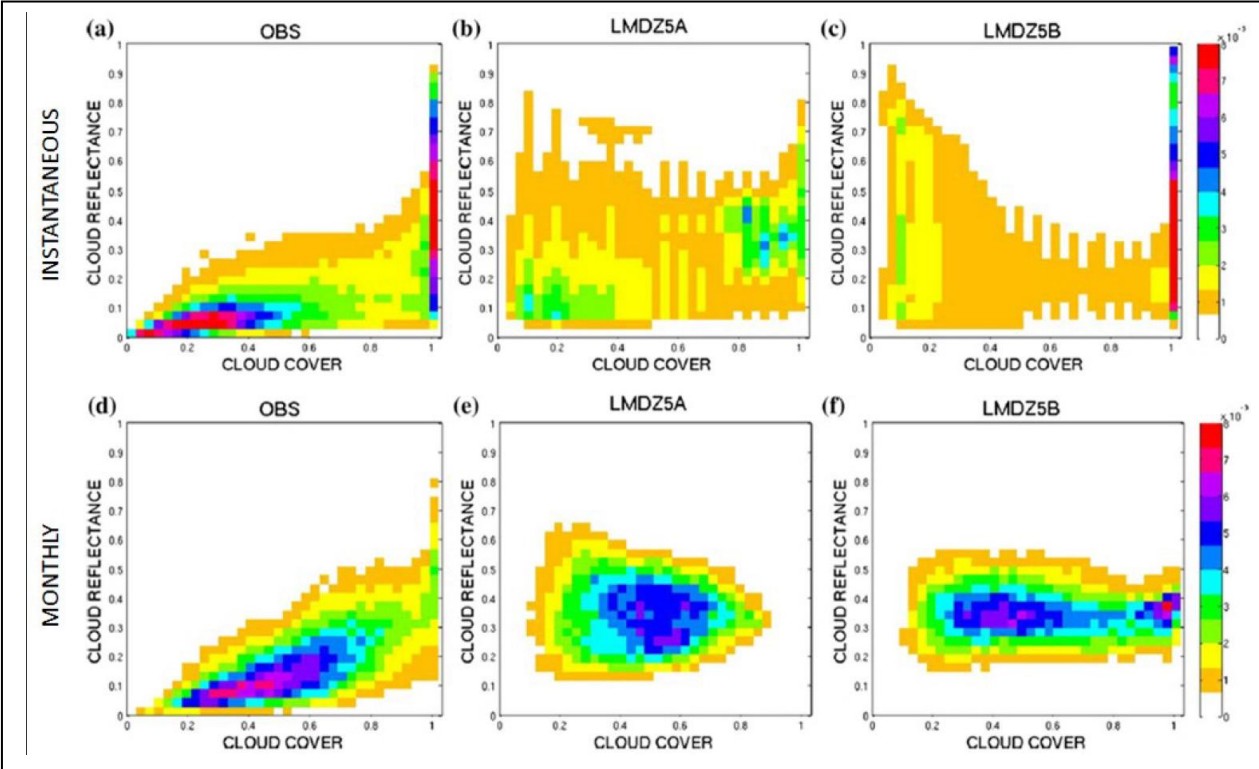

**Figure 8.** 2D histograms of cloud reflectance and cloud cover over the tropical oceans using instantaneous data (*upper panels)* and monthly data (*lower panels*) (a), (d) observed with PARASOL and CALIPSO-GOCCP, (b), (e) simulated with LMDZ5A and the simulator, and (c), (f) simulated with LMDZ5B and the simulator. The *colour bar* represents the number of points at each grid cell (cloud cover-cloud reflectance) divided by the total number of points. Reprinted with permission from Konsta et al. (2015).



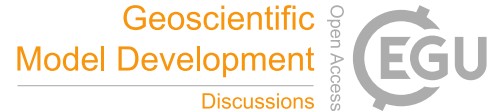

**Figure 9.** Comparison of the frequency of occurrence of clouds in the lowest 4 kms, of a given fraction at a given altitude under non-overlapped low-level cloud conditions for (a) stratocumulus regime and (b) shallow cumulus regime for CALIPSO-GOCCP and CMIP5 models. Maps show the frequency of occurrence of each regime derived from CALIPSO observations and ERA-Interim reanalysis. Reprinted from Nam et al. (2012).





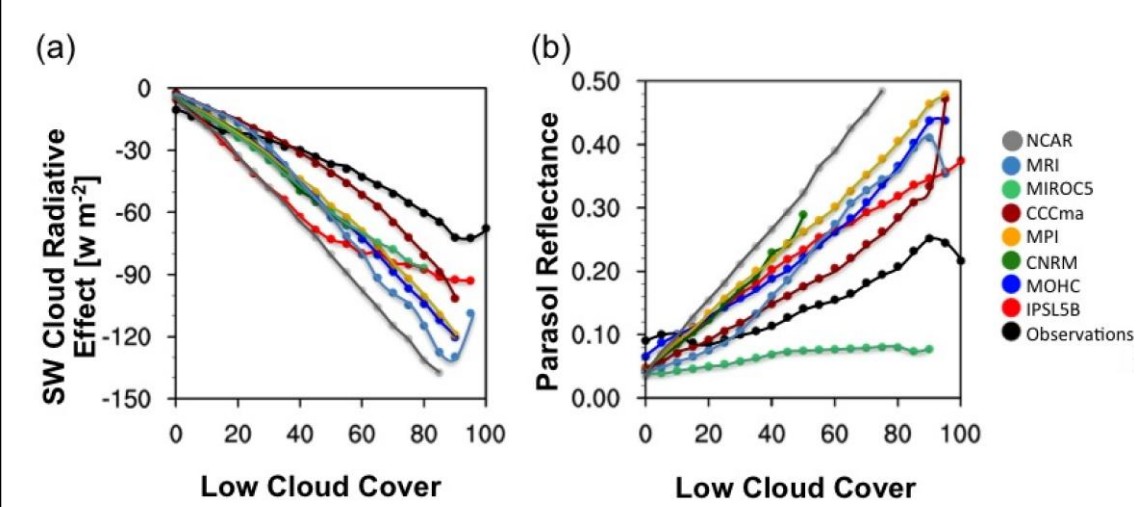

**Figure 10.** Mean relationship between non-overlapped low-cloud cover [%] and (a) the short-wave Cloud Radiative Effect [W m$^{-2}$]; and (b) the Parasol reflectance, derived for observations and for CMIP5 models over the Tropical oceans (30°N–30°S) from June 2006 to December 2008. Black: CERES and Parasol observations respectively; Gray: NCAR; Light Blue: MRI; Light Green: MIROC5; Light Red: IPSL5B; Dark Blue: MOHC; Dark Green: CNRM; Dark Red: CCCma; Orange: MPI. (Redrawn with modification from Nam et al. (2012).





**Figure 11.** Examples for the CFODD statistics obtained from CloudSat and MODIS satellite observations (upper), UKMO/HadGEM2 (middle) and GFDL/CM3 (bottom) reproduced with data of Suzuki et al. (2015). The colour shading shows the probability density function of radar reflectivity [%dBZ$^{-1}$] normalized at each in-cloud optical depth. The statistics are classified according to the cloud-top effective particle radius (left to right).





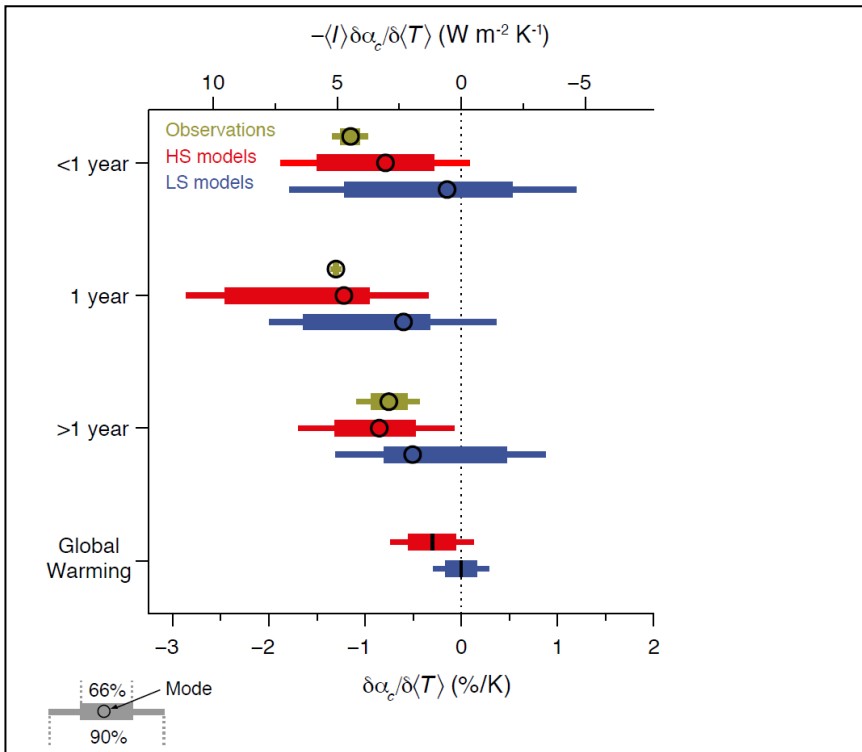

**Figure 12.** Observed and simulated covariance of TLC reflection with surface temperature. 29 CMIP5 models are used. Intra-annual (<1 yr), seasonal (1 yr), and interannual (>1 yr) frequency bands are distinguished. The regression coefficients $\delta\alpha_c/\delta\langle T\rangle$ are shown with their modes (most likely values) and 66% and 90% confidence intervals, for observations, 14 HS climate models, and 15 LS GCMs. Angle brackets $\langle\cdot\rangle$ denote the mean over the TLC regions. For the models, $\delta\alpha_c/\delta\langle T\rangle$ is also shown for global-warming simulations, calculated from the cloud reflection and temperature differences in the TLC regions between years 130–149 and years 2–11 after an abrupt quadrupling of carbon dioxide concentrations. For the global-warming simulations, the corresponding approximate confidence intervals ($0.95\sigma$ and $1.65\sigma$) obtained from the standard deviation $\sigma$ of $\delta\alpha_c/\delta\langle T\rangle$ among the HS and LS models are shown, with the bar marking the multimodel median. The upper axis indicates $-\langle I\rangle\delta\alpha_c/\delta\langle T\rangle$, which approximates the variation of the shortwave cloud radiative effect ($S_c$) with temperature, $\delta\langle S_c\rangle/\delta\langle T\rangle$. $\langle I\rangle$ is the region mean solar insolation. Reprinted from Brient and Schneider, 2016©American Meteorological Society. Used with permission.





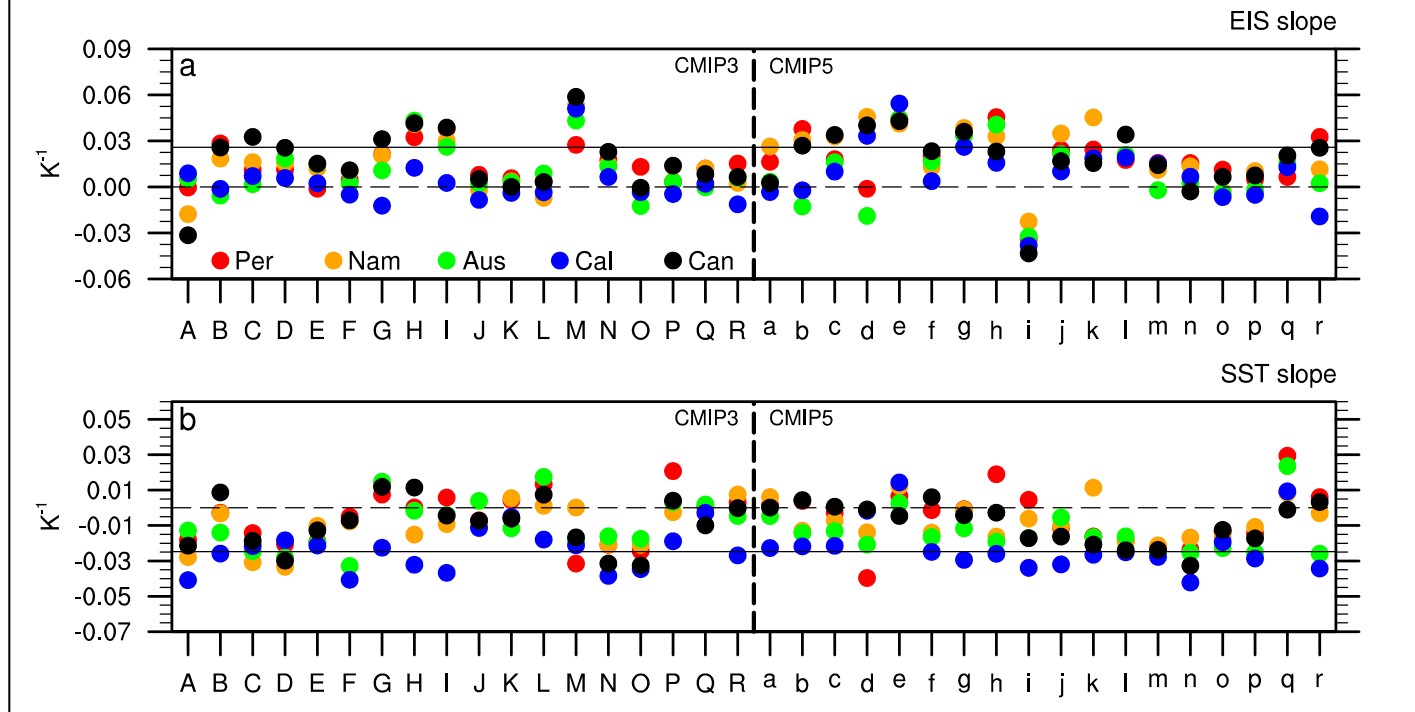

**Figure 13.** (a) The EIS slope($\partial LCC/\partial EIS$), (b) the SST slope ($\partial LCC/\partial SST$) in the 5 oceanic regions from 36 models in the 20th-century and from the observations. Note that the observational slope values (solid lines) are the averages over the 5 regions. Reprinted with permission from Qu et al. (2014).




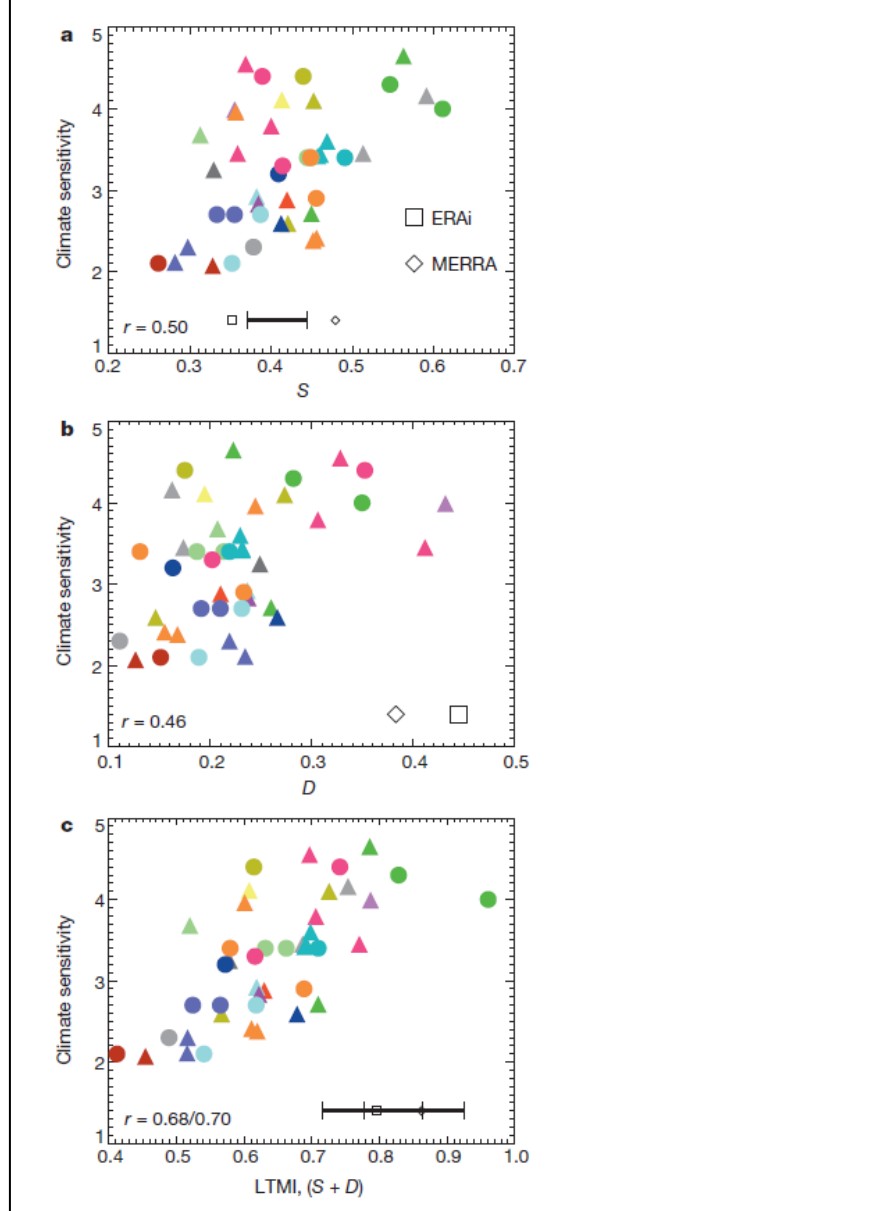

**Figure 14.** Scatterplot of S (a), D (b) and LTMI (the sum of S and D) (c) on the abscissa and the equilibrium climate sensitivity (on the ordinate) from 43 CMIP3 (circles) and CMIP5 models (triangles). Symbol color identifies modeling center of origin. Linear correlation coefficients are given in the lower left corner of LTMI with the equilibrium climate sensitivity and the total system feedback, respectively. Two observational estimates for LTMI with error bars are shown on the abscissa with central values indicated by the unfilled square and diamond. Reprinted from Sherwood et al. (2014).



**Figure 15.** Relative Frequency of Occurrence of ISCCP derived cloud regimes composited on a cyclone-centred reference framework (left) obtained from ERA-40-daily-mean sea level pressure and (right) for a model (GA2.0) 's cloud regimes and mean sea level pressure. In (g), the thick contours show the mean sea level pressure with 8hPa intervals. (h) in the left panel is a schematic with the typical position of the fronts in the cyclone composite. Reprinted from Bodas-Salcedo et al. (2015) ©American Meteorological Society. Used with permission.



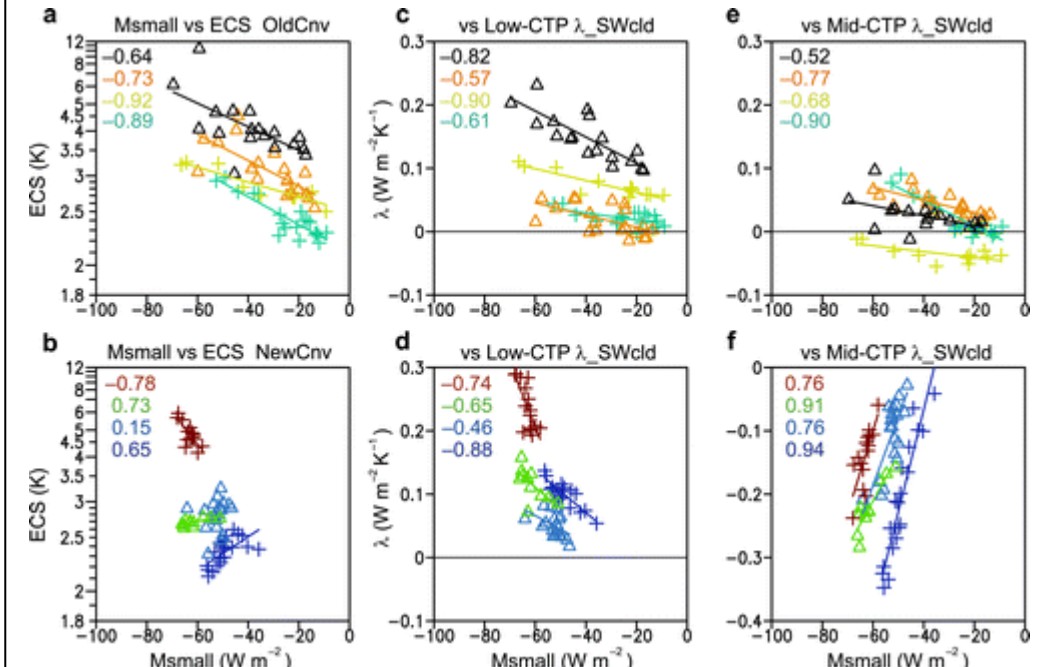

**Figure 16.** (a) Scatterplot of $M_{small}$ ($W\ m^{-2}$) (small-scale mixing) and ECS (K) for ensembles with an old convective scheme (OldCnv) subset. Values at top left in the panels indicate correlation coefficients of the individual perturbed physics ensembles (PPEs); (c) $M_{small}$ and shortwave cloud feedback parameter $\lambda_{SWcld}$ ($W\ m^{-2}\ K^{-1}$) in bins of 1.3–23 for $\tau$ and 800–1000 hPa for Cloud top pressure (CTP); and (e) $M_{small}$ and $\lambda_{SWcld}$ in bins of 23–60 for $\tau$ and 440–680 hPa for CTP. (b), (d), (f) as in (a), (c), (e), but for ensembles with a new convective scheme (NewCnv) subset. Reprinted from Kamae et al. (2016) ©American Meteorological Society. Used with permission.