# Peer review of "The Cloud Feedback Model Intercomparison Project (CFMIP) Diagnostic Codes Catalogue diagnostics metrics. and \_ evaluate. understand methodologies to and improve the representation of clouds and cloud feedbacks in climate models"

_Geoscientific Model Development, 2017_

## Referee Comment (RC1) · Anonymous Referee #1 · 21 Jun 2017

The manuscript organizes and briefly describes various diagnostics, together with programming codes, which are currently available in the CFMIP diagnostic code catalogue. The diagnostics can be applied to climate model output to evaluate model performance, to understand representation of key physical processes that contribute to model differences, and improve the representation of clouds and cloud feedbacks in

climate models. Having various diagnostics approaches catalogued and maintained in a single repository is extremely helpful and is a valuable contribution to the research community. Thus, I recommend acceptance with a few minor revisions listed below.

*It would be helpful if the authors provided sample input data and output results so that if the code needed to be re-written in another programming language it would be easier to reproduce the original output results.

*It would be helpful if the authors included a table in the beginning of Section 3, which would display some details of each diagnostics approach described in the article (i.e. Why that diagnostic was useful, what input data it required, etc).

*For one of the diagnostics included in the article, I found that, in the original paper (Sherwood at el., 2014) the authors restricted measurements to tropical ocean regions from 160 W to 30 E to compute the D parameter. However, in the code provided in GitHub repository in support of the article, the authors use area between 160 W and 45 E to compute the D. I would recommend keeping the method used in the code consistent with the original research.

*Figure 3: I think this Figure is unnecessary and I suggest removing it.

*Figure 8: I have difficulty reading the text associated with the colorbars.

---

## Referee Comment (RC2) · Anonymous Referee #2 · 1 Jul 2017

This manuscript presents an ensemble of up to date diagnostics to evaluate the clouds as simulated by atmospheric models, in particular for those who participate to the CMIP project. This initiative is very welcome and should be encouraged. The availability of these diagnostics will facilitate their use within the community.

However, the usability of these diagnostics are very different among them. Some di-

agnostics provide both the code and an ensemble of auxiliary files (e.g. diagnostic presented in sect. 3.1, 3.3, etc.), but many of them use make use of auxiliary files that do not exist in the repository (e.g. diagnostic presented in sect. 3.4, 3.5, 3.6, etc.). It will be very useful to have all the necessary files to be able to run a small demo for each diagnostic. As the best is the enemy of the good, I encourage the authors to be give as much information and files as possible, and to classify their diagnostics in a few categories, from the diagnostic that have all the necessary file to run a small demo to diagnostics that will require a lot of preprocessing and auxiliary material.

I recommend to accept this paper after modifications to improve it.

Detailed comments:

- The list of input and output variables are missing for some diagnostics (e.g. 3.7, 3.10, ...)

- general: most of the diagnostics use monthly mean data, some of them use daily (or other frequency) data. Please provide some information.

- p. 5, line 5: the metrics also often quantify the distance with observations.

- section 3.1: MODIS data are used according to Fig 1, but are not mentioned in the text

- p. 4 l. 15: "measures of fidelity": can you say some words on it?

- p. 5, l. 13: specify unit of $a_i$ (W/m2)

- p. 5, sect 3.2.2: this diagnostic is not available on github

- p. 5, l. 20-24: the explanation is not clear for me.

- sect 3.4, 3.6 and 3.7: are the ncl routines loaded at the beginning of the srcipt standard routines?

- p. 7, sect. 3.5: the matlab routines use a large number of auxiliary files

[Figure]

[Figure]

- p. 7, l. 18: "and the cloud vertical distribution from CALIPSO": not on the figures, should be removed

- p. 12: line 9 and followings: "The too few, too bright problem." The Konsta et al., 2015 paper should also be cited here.

- p. 12, l. 27: CFODDs: please expand the acronym

---

## Author Comment (AC2) · 28 Jul 2017

Thank you for your comments, which were useful to improve our repository and paper. Please see our responses below.

>To give as much information and files as possible, and to classify their diagnostics in a

[Figure]

few categories, from the diagnostic that have all the necessary file to run a small demo to diagnostics that will require a lot of preprocessing and auxiliary material.

Thank you for your suggestion. We have now included the corresponding information in each repository. If the sample input data size exceeds the GitHub limit, either the location information is provided or it is suggested to contact the author, which is described in the Readme page for each repository. We also have created a table which summarizes the descriptions of each of the diagnostics. (Attached as a figure file)

>The list of input and output variables are missing for some diagnostics (e.g. 3.7, 3.10,...)

All diagnostic repositories were supposed to provide information about the input data required and the output variables in the Readme pages for their repositories. We apologise for the missing information for some of the diagnostics. We have now added the list of input and output variables for all diagnostics. For 3.10 the author was extremely busy, I created a folk branch, added these information to its Readme file. Pull request has been sent to the author.

>general: most of the diagnostics use monthly mean data, some of them use daily (or other frequency) data. Please provide some information.

The information is included in the new summary table.

>p. 5, line 5: the metrics also often quantify the distance with observations.

The sentence is amended as follows: '), $m\_i$ is the error in simulating the regime i, which quantifies the distance from the observations, as defined below.'

>section 3.1: MODIS data are used according to Fig 1, but are not mentioned in the Text

The following sentences have been added to the manuscript: "As a point of comparison, we also use roughly analogous observations from the MODerate resolution Imaging Spectrometer (MODIS) instruments for the period March 2000 through April 2011 [Pincus et al., 2012].In Fig 1 (a), the ETCA measure between the MODIS and ISCCP climatologies is 0.47. All model differences with ISCCP exceed this value, so it is likely that errors in the climatology of total cloud amount are robustly determined."

>p. 4 l. 15: "measures of fidelity": can you say some words on it?

We have added 'closeness to the observations (fidelity)' where the word 'fidelity' appears for the first time in the manuscript.

>p. 5, l. 13: specify unit of a_i (W/m2)

It is specified as follows: "The model error (RMSE) associated with each regime i (a_i[Wm-2]) can be approximated. . ."

>p. 5, sect 3.2.2: this diagnostic is not available on github

There was a mistake with adding this diagnostic to the repository. The diagnostic code is now under the repository: https://github.com/tsussi/cloud-regime-error-metric/tree/master/code/

>p. 5, l. 20-24: the explanation is not clear for me.

The explanation has been modified (a figure file attached):

>sect 3.4, 3.6 and 3.7: are the ncl routines loaded at the beginning of the srcipt standard routines?

Yes, the NCL routines loaded are standard routines which are used in all scripts.

>p. 7, sect. 3.5: the matlab routines use a large number of auxiliary files

We have modified the matlab routine. The auxiliary files that are necessary give information on the model grids and on the choice of the appropriate solar zenith angle. We have added explicit information on how to obtain the auxiliary data in the Preprocessing section.

>p. 7, l. 18: "and the cloud vertical distribution from CALIPSO": not on the figures, should be removed

The sentence was amended to "the histogram shows the relationship between cloud cover from CALIPSO (Winker et al., 2007) and cloud reflectance measured by PARA-SOL (Parol et al., 2004)"

>p. 12: line 9 and followings: "The too few, too bright problem." The Konsta et al., 2015 paper should also be cited here.

Following sentences have been added: "Konsta et al., (2015) confirmed this at the instantaneous time scale. The tropical low-level cloud properties are grouped in two clusters according to the observations. One cluster corresponds to cumulus-type clouds (with low cloud fraction and low cloud reflectance), while the other corresponds to stratocumulus-type clouds (with almost overcast cloud fraction and with large cloud reflectance values). However, in two versions of the LMDZ climate model (Dufresne et al., 2013, Hourdin et al., 2013a, Hourdin et al., 2013b) these properties are not reproduced. The clouds with small cloud cover have too large reflectance values and clouds with a cover close to one are overestimated.

>p. 12, l. 27: CFODDs: please expand the acronym

The acronym has been defined in earlier section 3.7 as 'Contoured Frequency by Optical Depth Diagram (CFODD)'. So we use this acronym in the later part of the manuscript.
* * *
| Diagnostics | Scientific target to evaluate | Does the code read CMIP data? (i.e. no preprocessing ) | Time Frequency of Input Data | What auxiliary data is needed/ provided? |
|---|---|---|---|---|
| Klein et al.(2013) | ISCCP global cloud amounts | Y | Monthly | Processed Obs data/Y |
| Williams and Webb (2008), Tsushima et al.(2013) annual | Annual mean climatology of cloud regimes | Y | Daily | Processed Obs data/Y |
| Tsushima et al.(2013) seasonal | Climatological seasonal cycle of cloud regimes | Y | Daily | Processed Obs data/Y |
| Zelinka et al.(2012) | Cloud radiative kernels | Y | Monthly | Radiative Kernel /Y |
| Nam and Quaas (2012) | Zonal plots of GCM cloud and hydrometeor fraction | Y (post-processing done in script) | Monthly | N/A |
| Konsta et al (2015) | Instantaneous A-train cloud property | N | 8 hourly Or daily | N/A |
| Nam et al.(2012)vertical distribution | Vertical distribution of low-clouds | Y (post-processing done in script) | Monthly | N/A |
| Nam et al.(2012)albedo | SW CRE and Parasol reflectance of low-clouds | Y (post-processing done in script) | Monthly | N/A |
| Suzuki et al.(2015) | Warm rain microphysical process diagrams | N | 6 hourly | Processed Obs data/Y |
| Brient and Schneider (2016) | Sensitivity of tropical low-cloud reflection to SST at various time scales and the constraint to ECS | N | Monthly | Processed Obs data/Y |
| Qu et al.(2014) | Sensitivities of low cloud cover to EIS and SST | Y | Monthly | The observational estimates of EIS and SST slopes in the figures not included. |
| Sherwood et al.(2014) | Lower Tropospheric Mixing Indices | Y (,but IO routine not included) | Monthly | Land-sea mask/Y |

**Fig. 1.** Summary Table

This scalar metric evaluates variations of climatological monthly mean net CRE over the chosen number of cloud regimes.

An error in the climatological annual variation of the CRE for regime $i$ can be caused by an error in the amplitude of the variation and an error in the pattern (e.g. phase, shape) of the time variation. The centred RMS error of the climatological seasonal variation of the CRE for regime $(s_i)$ relative to the observations is expressed as:

$$s_i^2 = \left(\sigma_{i,m} - \sigma_{i.o}\right)^2 + 2\sigma_{i,m}\sigma_{i,o}(1 - R) \tag{3}$$

, where $\sigma_{i,o}$ and $\sigma_{i,m}$ denote the standard deviation of the climatological monthly mean of the observed and modelled CRE for a regime $i$ from the climatological annual mean, $R$ is the linear correlation coefficient between the anomaly (difference from annual mean) of the model and that of the observation over the 12 months of the seasonal cycle. We use these standard deviations as a measure of the amplitude in the seasonal variation. The error in the amplitude of the variation $(s_{i,amp})$ is defined by:

$$s_{i,amp} = \sigma_{i,m} - \sigma_{i,o} \tag{4}$$

The second term of $s_i$ is a covariance term between the observations and the model. We define the error in the pattern of the time variation $(s_{i,cov})$ as:

$$s_{i,cov} = \sqrt{2\sigma_{i,m}\sigma_{i,o}(1 - R)} \tag{5}$$

(See Tsushima et al., 2013 for details).

**Fig. 2.** Description of the Section 3.2.2

---

## Author Response (AR1)

Response to comments from Referee #1

Thank you for the comments. All of these comments were useful to improve our repository and paper. Please see our responses below.

>To provide sample input data and output results so that if the code needed to be re-written in another programming language it would be easier to reproduce the original output results.

Thank you for this very useful suggestion. We have now included sample input data and output results for each repositories. If the sample input data size exceeds the GitHub limit, either the location information is provided or it is suggested to contact the author, which is described in the Readme page for each repository.

>To include a table in the beginning of Section 3, which would display some details of each diagnostics approach described in the article (i.e. Why that diagnostic was useful, what input data it required, etc)

We have created a table which summarizes the descriptions of each of the diagnostics. (Attached at the bottom of the response)

>For one of the diagnostics included in the article, I found that, in the original paper(Sherwood at el., 2014) the authors restricted measurements to tropical ocean regionsfrom 160 W to 30 E to compute the D parameter. However, in the code provided inGitHub repository in support of the article, the authors use area between 160 W and45 E to compute the D. I would recommend keeping the method used in the codeconsistent with the original research.

Since this author was extremely busy, I created a folk branch, modified the region from 45E to 30E. Pull request has been sent to the author.

>Figure 3: I think this Figure is unnecessary and I suggest removing it.

The figure has been removed.

>Figure 8: I have difficulty reading the text associated with the colorbars.

In the revised figure the font size of the number next to the colour bar has been increased. (The revised figure attached)

Response to Comments from Referee #2

Thank you for your comments, which were useful to improve our repository and paper. Please see our responses below.

>To give as much information and files as possible, and to classify their diagnostics in a few categories, from the diagnostic that have all the necessary file to run a small demo to diagnostics that will require a lot of preprocessing and auxiliary material.

Thank you for your suggestion. We have now included the corresponding information in each repository.  If the sample input data size exceeds the GitHub limit, either the location information is provided or it is suggested to contact the author, which is described in the Readme page for each repository. We also have created a table which summarizes the descriptions of each of the diagnostics.  (Attached at the bottom of the response)

>The list of input and output variables are missing for some diagnostics (e.g. 3.7, 3.10,...)

All diagnostic repositories were supposed to provide information about the input data required and the output variables in the Readme pages for their repositories. We apologise for the missing information for some of the diagnostics. We have now added the list of input and output variables for all diagnostics. For 3.10 the author was extremely busy, I created a folk branch, added these information to its Readme file. Pull request has been sent to the author.

>general: most of the diagnostics use monthly mean data, some of them use daily (or other frequency) data. Please provide some information.

The information is included in the new summary table.

>p. 5, line 5: the metrics also often quantify the distance with observations.

The sentence is amended as follows: '), $m_i$ is the error in simulating the regime *i*, which quantifies the distance from the observations, as defined below.'

>section 3.1:  MODIS data are used according to Fig 1, but are not mentioned in the Text

The following sentences have been added to the manuscript:

> "As a point of comparison, we also use roughly analogous observations from the MODerate resolution Imaging Spectrometer (MODIS) instruments for the period March 2000 through April 2011 [Pincus et al., 2012].In Fig 1 (a), the $E_{TCA}$ measure between the MODIS and ISCCP climatologies is 0.47. All model differences with ISCCP exceed this value, so it is likely that errors in the climatology of total cloud amount are robustly determined."

>p. 4 l. 15: "measures of fidelity": can you say some words on it?

We have added 'closeness to the observations (fidelity)' where the word 'fidelity' appears for the first time in the manuscript.

>p. 5, l. 13: specify unit of a_i (W/m2)

It is specified as follows: "The model error (RMSE) associated with each regime $i$ ($a_i[Wm^{-2}]$) can be approximated…"

>p. 5, sect 3.2.2: this diagnostic is not available on github

There was a mistake with adding this diagnostic to the repository. The diagnostic code is now under the repository: https://github.com/tsussi/cloud-regime-error-metric/tree/master/code/

>p. 5, l. 20-24: the explanation is not clear for me.

The explanation has been modified as follows:

"This scalar metric evaluates variations of climatological monthly mean net CRE over the chosen number of cloud regimes.

An error in the climatological annual variation of the CRE for regime $i$ can be caused by an error in the amplitude of the variation and an error in the pattern (e.g. phase, shape) of the time variation. The centred RMS error of the climatological seasonal variation of the CRE for regime ($s_i$) relative to the observations is expressed as:

$$s_i^2 = (\sigma_{i,m} - \sigma_{i,o})^2 + 2\sigma_{i,m}\sigma_{i,o}(1-R) \qquad (3)$$

, where $\sigma_{i,o}$ and $\sigma_{i,m}$ denote the standard deviation of the climatological monthly mean of the observed and modelled CRE for a regime $i$ from the climatological annual mean, $R$ is the linear correlation coefficient between the anomaly (difference from annual mean) of the model and that of the observation over the 12 months of the seasonal cycle. We use these standard deviations as a measure of the amplitude in the seasonal variation. The error in the amplitude of the variation $(s_{i,amp})$ is defined by:

$$s_{i,amp} = \sigma_{i,m} - \sigma_{i,o} \qquad (4)$$

The second term of $s_i$ is a covariance term between the observations and the model. We define the error in the pattern of the time variation $(s_{i,cov})$ as:

$$s_{i,cov} = \sqrt{2\sigma_{i,m}\sigma_{i,o}(1-R)} \qquad (5)$$

(See Tsushima et al., 2013 for details). "

>sect 3.4, 3.6 and 3.7: are the ncl routines loaded at the beginning of the srcipt stan-

     dard routines?

Yes, the NCL routines loaded are standard routines which are used in all scripts.

>p. 7, sect. 3.5: the matlab routines use a large number of auxiliary files

We have modified the matlab routine. The auxiliary files that are necessary give information on the model grids and on the choice of the appropriate solar zenith angle. We have added explicit information on how to obtain the auxiliary data in the Preprocessing section.

>p. 7, l. 18: "and the cloud vertical distribution from CALIPSO": not on the figures, should be removed

The sentence was amended to "the histogram shows the relationship between cloud cover from CALIPSO (Winker et al., 2007) and cloud reflectance measured by PARASOL (Parol et al., 2004)"

>p. 12: line 9 and followings: "The too few, too bright problem." The Konsta et al., 2015

     paper should also be cited here.

Following sentences have been added: "Konsta et al., (2015) confirmed this at the instantaneous time scale. The tropical low-level cloud properties are grouped in two clusters according to the observations. One cluster corresponds to cumulus-type clouds (with low cloud fraction and low cloud reflectance), while the other corresponds to stratocumulus-type clouds (with almost overcast cloud fraction and with large cloud reflectance values). However, in two versions of the LMDZ climate model (Dufresne et al., 2013, Hourdin et al., 2013a, Hourdin et al., 2013b) these properties are not reproduced. The clouds with small cloud cover have too large reflectance values and clouds with a cover close to one are overestimated.

>p. 12, l. 27: CFODDs: please expand the acronym

The acronym has been defined in earlier section 3.7 as 'Contoured Frequency by Optical Depth Diagram (CFODD)'. So we use this acronym in the later part of the manuscript.

[revised manuscript text omitted]

---

## Author Response (AR2)

Response to comments from the topical editor

Thank you very much for your comments. Please see our responses below.

> 1.Repositories are evolving over time and new diagnostics may be added. Hence the overview page https://github.com/tsussi/cfmip-diagnostics-code-repository will probably change over time. To help future readers of the GMD article, it is mandatory to include a section where everyone will be able to access the code versions used in this paper and which will keep pointing to those versions, even in a few years from now.

We have updated the addresses in a supplementary file: List_versions.pdf , which contains the version of the codes at the time of the publication of this paper. We have also changed the address of the links, replacing the word 'commit' by 'tree'.  For example,

from: https://github.com/tsussi/cloud-regime-error-metric/commit/91dc2e368a8de111b5ab92b77edceeb12e171ed8

to:  https://github.com/tsussi/cloud-regime-error-metric/tree/91dc2e368a8de111b5ab92b77edceeb12e171ed8

 The new address shows the repository in a clearer layout to the user.

```
We have added a sentence in section 2 of the manuscript which informs the
reader that the list in the supplementary information contains links to all
GMD-documented diagnostics:
'The repositories are likely to evolve over time. To access the code versions
used in this paper, please see the supplementary information, where links to
all of the GMD-documented diagnostics are listed.'
```

We have also added this list to the github repository of the CFMIP diagnostics code repository and added a sentence near the bottom of the README page:

'To access the code versions used in the CFMIP diagnostics codes paper, please see List_versions.pdf, which contains links to all of the GMD-documented diagnostics.'

>2.Section 2 of the manuscript reads like an invitation to add their codes to the repository.

This should not be main purpose of this section. Instead you should describe the current status of the repository and give instructions on how to use it currently (and in the future, see above)

We have modified sentence s in the section as follows:

[revised manuscript text omitted]